# EMP-SSL: Towards Self-Supervised Learning in One Training Epoch

## Abstract

Recently, self-supervised learning (SSL) has achieved tremendous success in learning image representation. Despite the empirical success, most self-supervised learning methods are rather "inefficient" learners, typically taking hundreds of training epochs to fully converge. In this work, we show that the key towards efficient self-supervised learning is to increase the number of crops from each image instance. Leveraging one of the state-of-the-art SSL method, we introduce a **simplistic** form of self-supervised learning method called Extreme-Multi-Patch Self-Supervised-Learning (EMP-SSL) that does not rely on many heuristic techniques for SSL such as weight sharing between the branches, feature-wise normalization, output quantization, and stop gradient, etc, and reduces the training epochs by two orders of magnitude. We show that the proposed method is able to converge to 85.1% on CIFAR-10, 58.5% on CIFAR-100, 38.1% on Tiny ImageNet and 58.5% on ImageNet-100 in just **one** epoch. Furthermore, the proposed method achieves 91.5% on CIFAR-10, 70.1% on CIFAR-100, 51.5% on Tiny ImageNet and 78.9% on ImageNet-100 with linear probing in **less than ten** training epochs. In addition, we show that EMP-SSL shows significantly better transferability to out-of-domain datasets compared to baseline SSL methods.

## 1 Introduction

In the past few years, tremendous progress has been made in unsupervised and self-supervised learning (SSL) (LeCun, 2022). Classification performance of representations learned via SSL has even caught up with supervised learning or even surpassed the latter in some cases (Grill et al., 2020; Chen et al., 2020). This trend has opened up the possibility of large-scale data-driven unsupervised learning for vision tasks, similar to what have taken place in the field of natural language processing (Brown et al., 2020; Devlin et al., 2018).

A major branch of SSL methods is joint-embedding SSL methods (He et al., 2020; Chen et al., 2020; Zbontar et al., 2021; Bardes et al., 2021), which try to learn a representation invariant to augmentations of the the same image instance. These methods have two goals: (1) Representation of two different augmentations of the same image should be close; (2) The representation space shall not be a collapsed trivial one[1], i.e., the important geometric or stochastic structure of the data must be preserved. Many recent works (Chen et al., 2020; Grill et al., 2020; Zbontar et al., 2021; Bardes et al., 2021) have explored various strategies and different heuristics to attain these two properties, resulting in increasingly better performance.

Despite the good final performance of self-supervised learning, most of the SOTA SSL methods happen to be rather "inefficient" learners. For example, on CIFAR-10 (Krizhevsky et al., 2009), most methods would require at least 400 epochs to reach 90%, whereas supervised learning typically can reach 90% on CIFAR-10 within less than ten training epochs. The convergence efficiency gap is surprisingly large.

While the success of SSL has been demonstrated on a number of benchmarks, the principle or reason behind the success of this line of methods remains largely unknown. Recently, the work (Chen et al., 2022a) has revealed that the success of SOTA joint-embedding SSL methods can be explained by learning distributed representation of image patches, and this discovery echos with the

---

[1]For example, all representations collapse to the same point.

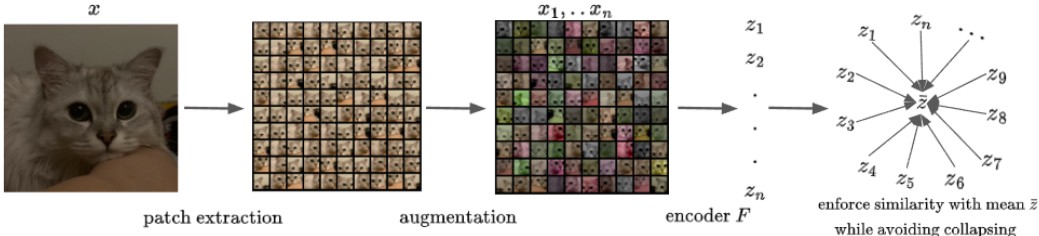

Figure 1: **The pipeline of the proposed method.** During the training, a image is randomly cropped into $n$ fixed-size image patches with overlapping. We then apply augmentation including color jitter, greyscale, horizontal flip, gaussian blur and solarization (Bardes et al., 2021) to $n$ fixed-size patches. Like other SSL methods (Chen et al., 2020; Bardes et al., 2021; Zbontar et al., 2021), image patches are then passed into the encoder $F$ to get the representations $z$.

discovery of BagNet (Brendel & Bethge, 2019) in the supervised learning regime. Specifically, the work (Chen et al., 2022a) show that joint-embedding SSL methods rely on successful learning the co-occurrence statistics of small image patches, and linearly aggregating of the patch representation as image representation leads to on-par or even better representation than the baseline methods. Similarly, another work based on sparse manifold transform (SMT) of small image patches (Chen et al., 2022b) has shown that simple white-box method can converge to close to SOTA performance in only *one* epoch. Given these observations, one natural question arises:

*Can we make self-supervised learning converge faster, even in one training epoch?*

In this work, we answer this question by leveraging the observation in (Chen et al., 2022a) and by pushing the number of crops in joint-embedding SSL methods to an extreme. We offer a new training paradigm called Extreme-Multi-Patch Self-Supervised Learning (EMP-SSL). With a simplistic formulation of joint-embedding self-supervised learning, we demonstrate that the SSL training epochs can be reduced by about **two orders of magnitude**. In particular, we show that EMP-SSL can achieve 85.1% on CIFAR-10, 58.5% on CIFAR-100, 38.1% on Tiny ImageNet and 58.5% on ImageNet-100 in just **one** training epoch. Moreover, with linear probing and a standard ResNet-18 backbone (He et al., 2016), EMP-SSL achieves 91.5% accuracy on CIFAR-10, 70.1% on CIFAR-100, 51.5% on Tiny ImageNet, and 78.9% on ImageNet-100 in less than ten training epochs. Remarkably, EMP-SSL achieves benchmark performance similar to that of SOTA methods, with more than two orders of magnitude less training epochs.

## 2 THE EXTREME-MULTI-PATCH SSL FORMULATION

**The Overall Pipeline.**   Like other methods in SSL (Chen et al., 2020; 2022a; Bardes et al., 2021; Zbontar et al., 2021), EMP-SSL operates on a joint embedding of augmented views of images. Inspired by the observation in (Chen et al., 2022a), the augmented views in EMP-SSL are fixed-size image patches with augmentation. As discussed in the previous section, the purpose of joint-embedding self-supervised learning is to enforce different image patches from the same image to be close while avoiding collapsed representation. The success of these methods comes from learning patch co-occurrence (Chen et al., 2022a). In order to learn the patch co-occurrence more efficiently, we increase the number of patches in self-supervised learning to an extreme.

For a given image $x$, we divide it into $n$ fixed-size patches using random crops with overlap. Each of these cropped patches undergoes standard augmentation same as those in VICReg (Bardes et al., 2021), resulting in augmented image patches $x_1, ..., x_n$. We denote $x_i$ as the $i$-th augmented image patch from $x$. For an augmented image patch $x_i$, we get embedding $h_i$ and projection $z_i$, where $h_i = f(x_i; \theta)$ and $z_i = g(h_i; \eta)$. At last, we normalize the projection $z_i$ learned. The parameter function $f(\cdot; \theta)$ is a deep neural network (ResNet-18 for example) with parameters $\theta$ and $g(\cdot; \eta)$ is a much simpler neural network with only two fully connected layers. We define our encoder $F$ as $F = g(f(\cdot; \theta); \eta)$. The pipeline is illustrated as Figure 1.

During the training, for a batch of $b$ images we denote as $X = [x^1, ..., x^b]$, where $x^j$ is the $j$-th image in the batch. We first augment the images as described above to get $X_1, .., X_n$ where

$X_i = [x_i^1, .., x_i^b]$. Then, we pass the augmented image patches into the encoder to get the features $Z_i = F(X_i)$ and concatenate them into $Z = [Z_1, ..., Z_n]$.

In this work, we adopt Total Coding Rate (TCR) (Ma et al., 2007; Li et al., 2022; Yu et al., 2020; Dai et al., 2022), which is a covariance regularization technique, to avoid collapsed representation:

$$R(Z) = \frac{1}{2} \log \det \left( I + \frac{d}{b\epsilon^2} Z Z^\top \right), \tag{1}$$

where $b$ is the batch size, $\epsilon$ is a chosen size of distortion with $\epsilon > 0$, and $d$ is the dimension of projection vectors. It can be seen as a soft-constrained regularization of covariance term in VICReg (Bardes et al., 2021), where the covariance regularization is achieved by maximizing the Total Coding Rate (TCR).

We aim for representations of different patches from the same image to be consistent, ensuring they're close in the representation space. This involves reducing the distance between the representation of augmented images and their mean patch representations. The training objective is:

$$\max \quad \frac{1}{n} \sum_{i=1,...,n} \left( R(Z_i) + \lambda D(Z_i, \bar{Z}) \right), \tag{2}$$

where $\lambda$ is the weight for invariance loss and $\bar{Z} = \frac{1}{n} \sum_{i=1,...,n} Z_i$ is the mean of representations of different augmented patches. In this work, we choose Cosine Similarity to implement the Distance function $D$, where $D(Z_1, Z_2) = Tr(Z_1^T Z_2)$ Hence, the larger value of $D$, the more similar $Z_i$ is to $\bar{Z}$. The pseudocode for EMP-SSL is shown as Algorithm 1 in the Appendix.

The objective equation 2 can be seen as a variant to the maximal rate reduction objective (Yu et al., 2020), or a generalized version of many covariance-based SSL methods such as VICReg (Bardes et al., 2021), I$^2$-VICReg (Chen et al., 2022a), TCR (Li et al., 2022) and Barlow Twins (Zbontar et al., 2021), in which $n$ is set to 2 for the common 2-view self-supervised learning methods. In this work, we choose $n$ to be much larger in order to learn the co-occurrence between patches much faster. Details can be found in Section 3.

**Bag-of-Feature Model.** Similar to (Chen et al., 2022a; Li et al., 2022), we define the representation of a given image $x$ to be the average of the embedding $h_1, ..., h_n$ of all the image patches. It is argued by (Chen et al., 2022a; Appalaraju et al., 2020) that the representation on the embedding $h_i$ contains more equivariance and locality that lead to better performance, whereas the projection $z_i$ is more invariant. An experimental justification can be found in (Appalaraju et al., 2020; Chen et al., 2022a), while a rigorous justification remains an open problem.

**Architecture.** In this work, we adopt the simplistic form of network architecture used in self-supervised learning. Specifically, EMP-SSL does not require prediction networks, momentum encoders, non-differentiable operators, or stop gradients. While these methods have been shown to be effective in some self-supervised learning approaches, we leave their exploration to future work. Our focus in this work is to demonstrate the effectiveness of a simplistic yet powerful approach to self-supervised learning.

## 3 EMPIRICAL RESULTS

In this section, we first verify the efficiency of the proposed objective in terms of convergence speed on standard datasets: CIFAR-10 (Krizhevsky et al., 2009), CIFAR-100 (Krizhevsky et al., 2009), Tiny ImageNet (Le & Yang, 2015) and ImageNet-100 (Deng et al., 2009). We then use t-SNE maps to show that, despite only a few epochs, EMP-SSL already learns meaningful representations. Next, we provide an ablation study on the number of patches $n$ in the objective equation 2 to justify the significance of patches in the convergence of our method. Finally, we present some empirical observations that the proposed method enjoys much better transferability to out-of-distribution datasets compared with other SOTA SSL methods.

**Experiment Settings and Datasets.** We provide empirical results on the standard CIFAR-10(Krizhevsky et al., 2009), CIFAR-100 (Krizhevsky et al., 2009), Tiny ImageNet (Le & Yang,

2015) and ImageNet-100 (Deng et al., 2009) datasets, which contains 10, 100, 200 and 100 classes respectively. Both CIFAR-10 and CIFAR-100 contain 50000 training images and 10000 test images, size $32 \times 32 \times 3$. Tiny ImageNet contains 200 classes, 100000 training images and 10000 test images. Image size of Tiny ImageNet is $64 \times 64 \times 3$. ImageNet-100 is a common subset of ImageNet with 100 classes [2], containing around 126600 training images and 5000 test images, size $224 \times 224$.

For all the experiments, we use a ResNet-18 (He et al., 2016) as the backbone and train for at most 30 epochs. We use a batch size of 100, the LARS optimizer (You et al., 2017) with $\eta$ set to 0.005, and a weight decay of 1e-4. The learning rate is set to 0.3 and follows a cosine decay schedule with a final value 0. In the TCR loss, $\lambda$ is set to 200.0 and $\epsilon^2$ is set to 0.2. The projector network consists of 2 linear layers with respectively 4096 hidden units and 512 output units. The data augmentations used are identical to those of VICReg (Bardes et al., 2021). For the number of image patches, we have set $n$ to 200 unless specified otherwise. For both CIFAR-10 and CIFAR-100, we use fixed-size image patches $16 \times 16$ and upsample to $32 \times 32$. For Tiny ImageNet, we use a fixed patch size of $32 \times 32$ and upsample to $64 \times 64$ for the convenience of using ResNet-18. For ImageNet-100, we use a fixed patch size of $112 \times 112$ and upsample to $224 \times 224$. We train an additional linear classifier to evaluate the performance of the learned representation. The additional classifier is trained with 100 epochs, optimized by SGD optimizer (Robbins & Monro, 1951) with a learning rate of 0.03.

**A Note on Reproducing Results of SOTA Methods.** We have selected five representative SOTA SSL methods (Chen et al., 2020; Grill et al., 2020; Bardes et al., 2021; Li et al., 2021; Caron et al., 2020) as baselines. For reproduction of other methods, we use sololearn (da Costa et al., 2022), which is one of the best SSL libraries on github. For CIFAR-10 and CIFAR-100, we run each method 3 times for 1000 epochs with their *optimal* parameters provided. For Tiny ImageNet, We notice that sololearn (da Costa et al., 2022) does not contain code to reproduce results on Tiny ImageNet and nearly all SOTA methods does not have official github code on Tiny ImageNet. So for fairness comparison, we adopt result from other peer-reviewed works (Ermolov et al., 2021; Zheng et al., 2021), in which SOTA methods are trained to 1000 epochs on ResNet-18. For ImageNet-100, we adopt results from sololearn (da Costa et al., 2022). All baseline methods run for 400 epochs, which is commonly used for these SSL methods.

Because our models are trained only on fixed-size image patches, we use bag-of-feature as the representation as described in Section 2. Following (Chen et al., 2022a), we choose 128 as the number of patches in the bag-of-feature. The other reproduced models follow the routine in (Chen et al., 2020; He et al., 2020; Bardes et al., 2021) and evaluate on the whole image. We acknowledge that this may give a slight advantage to EMP-SSL. But as shown in Table 1, 2, 3 in (Chen et al., 2022a), the difference between bag-of-feature and whole image evaluation in (Chen et al., 2020; He et al., 2020; Bardes et al., 2021) is at most 1.5%. We consider it negligible since this is a work about data efficiency of SSL methods, not about advancing the SOTA performance.

| Methods | CIFAR-10 1000 Epoch | CIFAR-100 1000 Epoch | Tiny ImageNet 1000 epochs | ImageNet-100 400 epochs |
|---|---|---|---|---|
| SimCLR | 0.910 | 0.662 | 0.488 | 0.776 |
| BYOL | 0.926 | 0.708 | 0.510 | 0.802 |
| VICReg | 0.921 | 0.685 | - | 0.792 |
| SwAV | 0.923 | 0.658 | - | 0.740 |
| ReSSL | 0.914 | 0.674 | - | 0.769 |
| EMP-SSL (**1 Epoch**) | 0.851 | 0.585 | 0.381 | 0.585 |

Table 1: **Performance of EMP-SSL with 1 epoch vs standard self-supervised SOTA methods converged.** Accuracy is measured by linear probing.

### 3.1 SELF-SUPERVISED LEARNING IN ONE EPOCH

We conducted an experiment for one epoch and set the learning rate weight decay to one epoch, while keeping all other experiment settings the same. Table 1 shows the results of our method, as

---

[2]The selection of 100 classes can be found in (da Costa et al., 2022).

well as some representative state-of-the-art (SOTA) SSL methods. From the Table, we observe that, even only seen the data once, the method is able to converge to a result close to the fully converged SOTA performance. This demonstrates great potential not only in improving the convergence of current SSL methods, but also in other fields where the data can only be seen once, such as in online learning, incremental learning and robot learning.

## 3.2 FAST CONVERGENCE ON STANDARD DATASETS

| Methods | CIFAR-10 | | | | CIFAR-100 | | | |
|---|---|---|---|---|---|---|---|---|
| | 1 Epoch | 10 Epochs | 30 Epochs | 1000 Epochs | 1 Epoch | 10 Epochs | 30 Epochs | 1000 Epochs |
| SimCLR | 0.282 | 0.565 | 0.663 | 0.910 | 0.054 | 0.185 | 0.341 | 0.662 |
| BYOL | 0.249 | 0.489 | 0.684 | **0.926** | 0.043 | 0.150 | 0.349 | **0.708** |
| VICReg | 0.406 | 0.697 | 0.781 | 0.921 | 0.079 | 0.319 | 0.479 | 0.685 |
| SwAV | 0.245 | 0.532 | 0.767 | 0.923 | 0.028 | 0.208 | 0.294 | 0.658 |
| ReSSL | 0.245 | 0.256 | 0.525 | 0.914 | 0.033 | 0.122 | 0.247 | 0.674 |
| EMP-SSL (20 patches) | **0.806** | **0.907** | **0.931** | - | **0.551** | **0.678** | **0.724** | - |
| EMP-SSL (200 patches) | **0.826** | **0.915** | **0.934** | - | **0.577** | **0.701** | **0.733** | - |

Table 2: **Performance on CIFAR-10 and CIFAR-100 of EMP-SSL and standard self-supervised SOTA methods in different epochs.** Accuracy is measured by training linear classifier on learned embedding representation. Since EMP-SSL already converges with 10 epochs, we do not run it to 1000 epochs like other SOTA methods. Best are marked in **bold**.

**Comparisons with Other SSL Methods on CIFAR-10 and CIFAR-100.** In Table 2, we present results of EMP-SSL trained up to 30 epochs and other SOTA methods trained up to 1000 epochs following the routine in (Chen et al., 2020; Bardes et al., 2021; Zbontar et al., 2021). On CIFAR-10, EMP-SSL is observed to converge much faster than traditional SSL methods. After just one epoch, it achieves 80.6% accuracy with 20 patches and 82.6% accuracy with 200 patches. In only ten epochs, it converges to more than 90%, which is considered as the state-of-the-art result for self-supervised learning methods on CIFAR-10. By 30 epochs, EMP-SSL surpasses all current methods, achieving over 93% accuracy as shown in the 1000 epochs column in Table 2. Similarly, EMP-SSL also converges very quickly on more complex datasets like CIFAR-100. In Table 2, with just 10 epochs, EMP-SSL is able to converge to 70.1% accuracy. The method further surpasses current SOTA methods with 30 epochs of training.

We also present EMP-SSL's plot of convergence on CIFAR-10 in Figure 2 and on CIFAR-100 in Figure 3, showcasing that EMP-SSL indeed converges very quickly. In particular, it only takes at most 5 epochs for the method to achieve over 90% on CIFAR-10 and over 65% on CIFAR-100 with 200 patches and at most 8 epochs with 20 patches. More importantly, it is evident EMP-SSL converges after 15 epochs on both datatsets, around 93% on CIFAR-10 and 72% on CIFAR-100.

**A Note on Time Efficiency.** It is admittedly true that increasing number of patches in joint-embedding self-supervised learning could lead to increased training time. Here, we compare the time needed for each method to reach a prescribed performance on CIFAR, 90% on CIFAR-10 and 65% on CIFAR-10. Results are in Table 3. On CIFAR-10, EMP-SSL not only requires far fewer training epochs to converge, but also less runtime. This advantage becomes more evident on more complicated CIFAR-100 dataset. While previous methods require more epochs and, therefore, longer time to converge, EMP-SSL uses a few epochs to reach a good result. This result provides empirical evidence that the proposed method would enjoy the faster speed of training, especially with the setting with 20 patches. Beyond advantage in efficiency, one may wonder how the model learned with a few epochs is different from previous methods learned with 1000 epochs. As we will further show in section 3.3 and 3.5, the so learned model possess its unique benefits.

**Comparisons with Other SSL Methods on Tiny ImageNet and ImageNet-100** We evaluated the performance of EMP-SSL on larger datasets, namely Tiny ImageNet and ImageNet-100. Table 4 presents the results of EMP-SSL trained for 10 epochs on these two datasets. Even on the more challenging dataset Tiny ImageNet, EMP-SSL is still able to achieve 51.5%, which is slightly better

| Methods | CIFAR-10 | | CIFAR-100 | |
|---|---|---|---|---|
| | Time | Epochs | Time | Epochs |
| SimCLR | 385 | 842 | 453 | 907 |
| BYOL | 142 | 310 | 171 | 320 |
| VICReg | 308 | 587 | 430 | 642 |
| SwAV | 162 | 150 | 264 | 241 |
| ReSSL | 194 | 447 | 211 | 488 |
| EMP-SSL (20 patches) | **35** | 8 | **30** | 7 |
| EMP-SSL (200 patches) | 142 | **5** | 112 | **4** |

Table 3: **Amount of time and epochs each method takes to reach 90% on CIFAR-10 and 65% on CIFAR-100.** Time is measured in minutes and best are marked in **bold**.

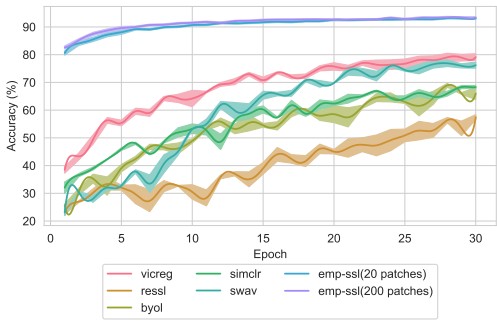
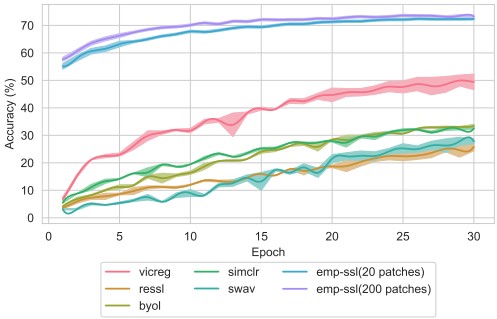

Figure 2: **The convergence plot of EMP-SSL trained on CIFAR-10 for 30 epochs.** The Accuracy is measured by linear probing. Each method runs 3 random seeds and standard deviation is displayed by shadows.

Figure 3: **The convergence plot of EMP-SSL trained on CIFAR-100 for 30 epochs.** The Accuracy is measured by linear probing. Each method runs 3 random seeds and standard deviation is displayed by shadows.

than SOTA methods trained with 1000 epochs. A similar result is observed on ImageNet-100. The method converges to the range SOTA performance within 10 epochs. The result shows the potential of our method in applying to data sets of larger scales.

## 3.3 VISUALIZING THE LEARNED REPRESENTATION

To further understand the representations learned by EMP-SSL with a few epochs, we visualize the features learned using t-SNE (Van der Maaten & Hinton, 2008). In Figure 4, we visualize the learned representations of the training set of CIFAR-10 by t-SNE. EMP-SSL is trained up to 10 epochs with 200 patches and other SOTA methods are trained up to 1000 epochs. All t-SNEs are produced with the same set of parameters. Each color represents one class in CIFAR-10. As shown in the figure, EMP-SSL learns much more separated and structured representations for different classes. Comparing to other SOTA methods, the features learned by EMP-SSL show more refined low-dim structures. For a number of classes, such as the pink, purple, and green classes, the method even learns well-structured representation inside each class. Moreover, the most amazing part is that all such structures are learned from training with just 10 epochs!

## 3.4 ABLATION STUDIES OF EMP-SSL

We provide ablation studies on the number of patches $n$ to illustrate the importance of patch number in joint-embedding SSL. All experiments on done on CIFAR-10, with training details same with the ones in 3. Figure 5 shows the effect that the number of patches $n$ has on the convergence and performance of EMP-SSL. As the number $n$ increases, the accuracy clearly rises sharply. Increasing number of patches $n$ used in training will facilitate the models to learn patch representation and the co-occurrence, and therefore accelerate the convergence of our model.

| Methods | Tiny ImageNet | | ImageNet-100 | |
|---|---|---|---|---|
| | Epochs | Accuracy | Epochs | Accuracy |
| SimCLR | 1000 | 0.488 | 400 | 0.776 |
| BYOL | 1000 | 0.510 | 400 | **0.802** |
| VICReg | - | - | 400 | 0.792 |
| SwAV | - | - | 400 | 0.740 |
| ReSSL | - | - | 400 | 0.769 |
| EMP-SSL (ours) | 10 | **0.515** | 10 | 0.789 |

Table 4: **Performance on Tiny ImageNet and ImageNet-100 of EMP-SSL vs SOTA SSL methods at different epochs**. Best results are marked in **bold**.

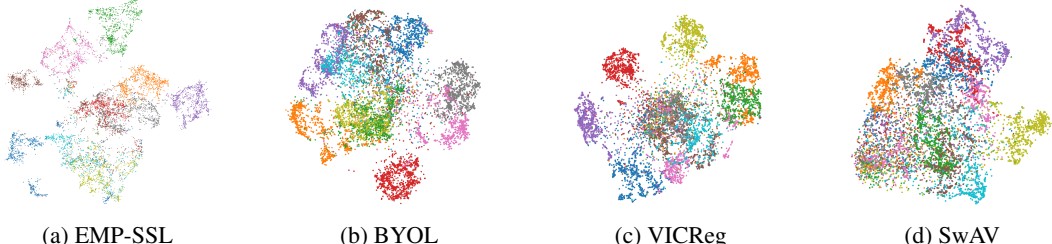

(a) EMP-SSL      (b) BYOL      (c) VICReg      (d) SwAV

Figure 4: **t-SNE of learned representation on CIFAR-10.** We use projection vectors to generate the t-SNE graph.

## 3.5 TRANSFERABILITY TO OUT OF DOMAIN DATA

Aside from converging with much fewer epochs, we are interested in whether EMP-SSL can bring additional benefits comparing to standard 2-view self-supervised learning methods trained to 1000 epochs. In this section, we provide an interesting empirical observation: the method's better transferability to out of domain data. We conduct two sets of experiments: (1) models pretrained on CIFAR-10 and linearly evaluated on CIFAR-100 (2) models pretrained on CIFAR-100 and linearly evaluated on CIFAR-10. We present the results of these two sets of experiments in Table 5. EMP-SSL is trained for 30 epochs and other self-supervised methods are trained for 1000 epochs like previous subsections. Note that despite similar names, CIFAR-10 and CIFAR-100 have very little overlap hence they are suitable for testing model's transferability.

| Methods | CIFAR-10 to CIFAR-100 | | CIFAR-100 to CIFAR-10 | |
|---|---|---|---|---|
| | In Domain | Out of Domain | In Domain | Out of Domain |
| SimCLR | 0.910 | 0.517 | 0.662 | 0.783 |
| BYOL | 0.926 | 0.552 | 0.708 | 0.813 |
| VICReg | 0.921 | 0.515 | 0.685 | 0.791 |
| SwAV | 0.923 | 0.508 | 0.658 | 0.771 |
| ReSSL | 0.914 | 0.529 | 0.674 | 0.780 |
| EMP-SSL (20 patch) | 0.931 | 0.645 | 0.724 | 0.857 |
| EMP-SSL (200 patch) | **0.934** | **0.648** | **0.733** | **0.859** |

Table 5: Transfer to out-of-domain data: We evaluate the model's transferability to out-of-domain data. Best results are in **bold**.

In Table 5, EMP-SSL clearly demonstrates better transferability to out of domain data comparing to models trained in 1000 epochs. Since the main goal of self-supervised learning is to develop data-driven machine learning on wide ranges of vision tasks, it is crucial for the self-supervised learning methods to generalize well to out-of-domain data instead of overfitting the training data. From the result shown in the table, we believe this work will help advance SSL methods in such a direction.

A potential reason for this phenomenon is that more training epochs lead to model overfitting. EMP-SSL, converging in fewer epochs, better avoids this issue. A detailed explanation is reserved for future studies.

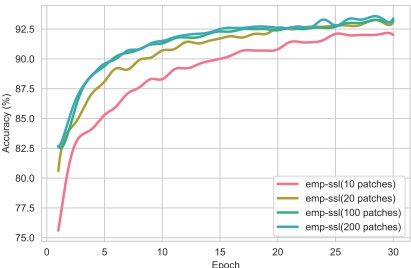

Figure 5: **Ablation Study on the number of patches** $n$**.** Experiments are conducted on CIFAR-10.

## 4 MORE RELATED WORKS

There are several intertwined quests closely related to this work. Here, we touch them briefly.

**Joint-Embedding Self-Supervised Learning.** Our work is mostly related to joint-embedding self-supervised learning. The idea of instance contrastive learning was first proposed in Wu et al. (2018) The method relies on a joint-embedding architecture in which two networks are trained to produce similar embeddings for different views of the same image. The idea can trace back to Siamese network architecture which was proposed in (Bromley et al., 1993). The main challenge to these methods is *collapse* where all representations are identical, ignoring the input. To overcome this issue, there are mainly two approaches: contrastive and information maximization. On the branch of contrastive learning, methods search for dissimilar samples from the current branch (Chen et al., 2020) or memory bank (He et al., 2020). More recently, a few methods jump out of the constraint of using contrastive samples. They exploit several tricks, such as the parameter vector of one branch being a low-pass-filtered version of the parameter vector of the other (Grill et al., 2020), stop-gradient operation in one branch (Chen & He, 2021) and batch normalization (Richemond et al., 2020).

On the other line of anti-collapse methods, several simpler non-constrastive methods are proposed to avoid the collapsed representation problem. TCR (Li et al., 2022), Barlow Twins (Zbontar et al., 2021), and VICReg (Bardes et al., 2021) propose covariance regularization to enforce a non-collapsing solution. Our work is constructed on the basis of covariance regularization to avoid collapsed representation.

Besides exploring ways to achieve anti-collapsing solution, SwAV (Caron et al., 2020) explores *multi-crop* in self-supervised learning. The work uses a mix of views with different resolutions in place of two full-resolution views. It is the first work to demonstrate that *multi-view* augmentation improves the performance of SSL learning. Our work simplifies and generalizes this approach and takes it to an extreme.

Aside from the empirical success of SSL learning, work like $I^2$-VICReg (Chen et al., 2022a) digs into the principle behind these methods. The work argues that success largely comes from learning a representation of image patches based on their co-occurrence statistics in the images. In this work, we adopt this observation and demonstrate that learning the co-occurrence statistics of image patches can lead to fundamental change in the efficiency of self-supervised learning as shown in Section 3.

**Patch-Based Representation Learning.** Our work is also closely related to representation learning on fixed-size patches in images. The idea of exploiting patch-level representation is first raised in the supervised setting. Bagnet (Brendel & Bethge, 2019) classifies an image based on the co-occurrences of small local image features without taking the spatial ordering into consideration. Note, this philosophy strongly echoes with the principle raised in (Chen et al., 2022a). The paper demonstrates that this "bag-of-feature" approach works very well on supervised classification tasks. Many follow-up works like SimplePatch (Thiry et al., 2021) and ConvMixer (Trockman & Kolter, 2022) have all demonstrated the power of patch representation in supervised learning.

In unsupervised learning, some early work like Jigsaw puzzle (Noroozi & Favaro, 2016) learns patch representation via solving a patch-wise jigsaw puzzle task and implicitly uses patch representation in self-supervised learning. Gidaris (Gidaris et al., 2020) takes the "bag-of-words" concept from NLP and applies it into the image self-supervision task. The work raises the concept of "bag-of-patches"

and demonstrates that this image discretization approach can be a very powerful self-supervision in the image domain. In the recent joint-embedding self-supervised domain, $I^2$-VICReg (Chen et al., 2022a) is the first work to highlight the importance of patch representation in self-supervised learning. There's another line of self-supervised learning work (Bao et al., 2021; He et al., 2022) based on vision transformers, which naturally uses fixed-size patch level representation due to the structure of the vision transformers.

**SSL Methods Not Based on Deep Learning.** Our work has also been inspired by the classical approaches before deep learning, especially sparse modeling and manifold learning. Some earlier works approach unsupervised learning mainly from the perspective of sparsity (Yu et al., 2009; Lazebnik et al., 2006; Perronnin et al., 2010). In particular, a work focuses on lossy coding (Ma et al., 2007) has inspired many of the recent SSL learning methods (Li et al., 2022; Chen et al., 2022a), as well as our work to promote covariance in the representation of data through maximizing the coding rate. Manifold learning (Hadsell et al., 2006; Roweis & Saul, 2000) and spectral clustering (Schiebinger et al., 2015; Meilă & Shi, 2001) propose to model the geometric structure of high dimensional objects in the signal space. In 2018, a work called sparse manifold transform (Chen et al., 2018) builds upon the above two areas. The work proposes to use sparsity to handle locality in the data space to build support and construct representations that assign similar values to similar points on the support. One may note that this work already shares a similar idea with the current joint-embedding self-supervised learning in the deep-learning community.

## 5 DISCUSSION

This paper seeks to solve the long-standing inefficient problem in self-supervised learning. We demonstrated that with an increased number of patches during training, the method of joint-embedding self-supervised can achieve a prescribed level of performance on various datasets, such as CIFAR-10, CIFAR-100, Tiny ImageNet, and ImageNet-100, in just one epoch. We show that the method further converges to the state-of-the-art performance in about ten epochs on these datasets. Despite converged with much fewer epochs, EMP-SSL not only learns meaningful representations but also shows advantages in tasks like transferring to out-of-domain datasets.

While the proposed method is simple and intuitive, it is unclear if it will work. Using many correlated patches may lead to catastrophic forgetting, as in continual learning. We show that it works nicely. This work makes one step towards efficient and online SSL. Our work does not propose another "new" SSL method per se, and it serves to show the key to efficiency is to leverage co-occurrence sufficiently.

This discovery opens the doors to many potential research, such as uncovering the mystery behind networks used in self-supervised learning and designing more interpretable and efficient "white-box" networks for learning in an unsupervised setting. This can potentially lead to more transparent and understandable models and advance the field of self-supervised learning in various applications.

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

## A    IMPLEMENTATION DETAILS

Due to the limited space in the main paragraph, we include a more detailed implementation of our method and reproduction of other methods in here.

### A.1    TRAINING DETAILS OF EMP-SSL

The augmentation used follows VICReg (Bardes et al., 2021). A pytorch stype pseudo code is listed below:

- transforms.RandomHorizontalFlip(p=0.5)
- transforms.RandomApply([transforms.ColorJitter(0.4, 0.4, 0.4, 0.2)], p=0.8)
- transforms.RandomGrayscale(p=0.2)
- GBlur(p=0.1)
- transforms.RandomApply([Solarization()], p=0.1)

All experiments are trained with at most 4 A100 GPUs.

**A note on reproducing our results.**    We think we have included very detailed descriptions to reproduce results of our method. Additionally, we are happy to share code with reviewer by request.

### A.2    TRAINING DETAILS OF OTHER METHODS

When reproducing methods of other work, we have adopted solo-Learn (da Costa et al., 2022) as described in the main paragraph. We followed the optimal parameters and augmentation provided by solo-learn. A special note is that we followed the default batch size, which is 256 because it is studied in many SSL methods (Chen et al., 2020; Bardes et al., 2021) that larger batch size will produce better performance.

## B    PSEUDO CODE FOR EMP-SSL

**Algorithm 1:**  EMP-SSL PyTorch Pseudocode

```
# F:  encoder network
# lambda:  weight on the invariance term
# n:  number of augmented fixed-size image patches
# m:  number of pairs to calculate invariance
# R:  function to calculate total coding rate
# D:  function to calculate cosine similarity
for X in loader:
   # augment n fixed-size image patches
   X_1...X_n = extract patches & augment(X)

   # calculate projection
   Z_1...Z_n = F(X_1)...F(X_n)

   # calculate total coding rate and invariance loss
   tcr_loss = average([R(Z_i) for i in range(n)]
   inv_loss = average([D(Z̄, Z_i) for i in range(n)])

   # calculate loss
   loss = tcr_loss + lambda*inv_loss

   # optimization step
   loss.backward()
   optimizer.step()
```

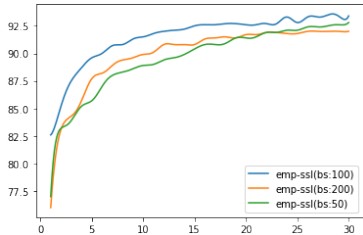

Figure 6: **Ablation Study on Batch Size** Experiments are conducted on CIFAR-10.

## C    MORE ABLATION STUDIES

In this section, we present more ablation studies of EMP-SSL.

### C.1    ABLATION ON BATCH SIZE

In this subsection, we verify if our method is applicable to different batch sizes. Again, we use CIFAR-10 to conduct ablation study and training details same in 3. We choose batch size of 50, 100, and 200 to conduct our ablation study. In all experiments, we use 200 patches and all the parameters are kept the same, in other words, we have not searched different hyperparameters for different batch sizes. We visualize the results of ablation study in Figure 6. One may observe that batch size has little impact on the convergence of EMP-SSL. The result is very important because different batch size leads to different iteration the method has run in the same epochs. It shows that, even without changing hyperparameters, the proposed method helps the convergence of SSL method under different batch sizes.

## D    T-SNE COMPARISON WITH OTHER METHODS

Due to limited space in the main text, we present the t-SNE of all of the SOTA SSL methods we have chosen to compare in here. We present the result of all t-SNE graphs in Figure 7. Here, we draw a similar conclusion as the main paragraph, that EMP-SSL learns highly structured representation in just 10 epochs.

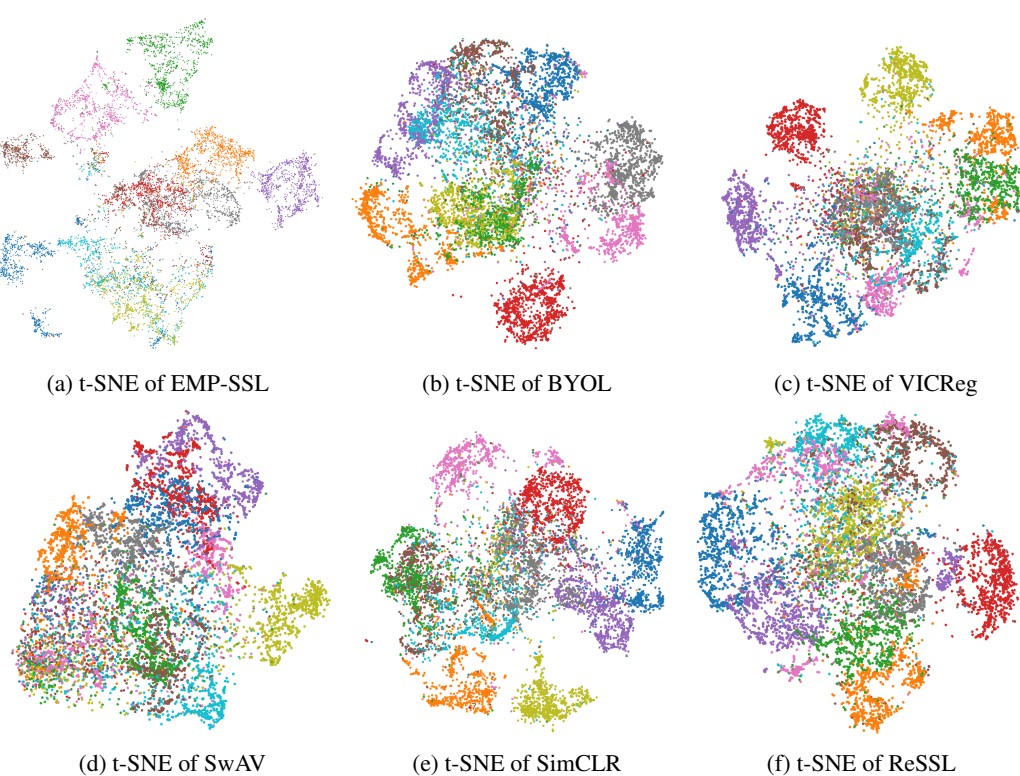

(a) t-SNE of EMP-SSL      (b) t-SNE of BYOL      (c) t-SNE of VICReg

(d) t-SNE of SwAV      (e) t-SNE of SimCLR      (f) t-SNE of ReSSL

Figure 7: **t-SNE of learned representation on CIFAR-10.** We use projection vectors trained on CIFAR-10 to generate the t-SNE graph.

