# OpenReview forum: "EMP-SSL: Towards Self-Supervised Learning in One Training Epoch"
_ICLR.cc/2024/Conference — ICLR 2024 Conference Withdrawn Submission_

### Official Review · Reviewer_R77R · 2023-10-27

**Soundness:** 2 fair
**Presentation:** 2 fair
**Contribution:** 2 fair
**Rating:** 3
**Confidence:** 5

**Summary:**

This work evaluates how an SSL strategy based on joint-embedding and multiple patches per image, aggregated through Bag-of-Features, achieve converge in small number of epochs. As training strategy, the authors propose a non-contrastive approach by minimizing the cosine distance between each patch's  representation and the average representation, as well as regularizing covariance through Total Coding Rate.

The experimental section focuses on showcasing the convergence speed, measured in number of epochs, compared to other SOTA methods on datasets like CIFAR-10/100, TinyImageNet and ImageNet-100. The authors additionally include ablation studies on the number of patches, and t-SNE visualizations.

**Strengths:**

* The overall goal of this paper is relevant, since training SSL in smaller time/compute is of bread interest.

* Several SOTA methods are used for comparison, which is good scientific practice.

**Weaknesses:**

* The overall novelty of the paper is unclear. This work uses Bag-of-Features (BoF) through a fixed-patch approach, and regularization via TCR. Both these strategies have already been explored, as the authors mention. As is, the paper lacks clarity with respect to its novelty.

* The comparison in "epochs" seems unfair to other methods. Efficiency in SSL should not be measured in terms of epochs, but rather in time/compute or some surrogate metric such as number of patches processed. I would also include a comparison in terms of GPU memory required per method.
  * Similarly, I could not find any discussion about the efficiency of EMP-SSL at test time.

* Although the method is compared to several other SOTA methods, the datasets used are rather small to extract general conclusions about SSL. Larger datasets (>= ImageNet-1K) would be required for that, specially because of the use of larger resolution images which might affect (positively or negatively, to be seen) the patching and BoF approach.

* I personally think the language in the paper could be improved. There are several informal expressions that could benefit from some rewriting. I also found several typos, I suggest the authors giving a careful read to polish those aspects, since language is an important aspect in the reading of a scientific paper.

**Questions:**

On the method:

* I kindly ask the authors to clearly expose the algorithmic novelty of this work, since it is not clear form the manuscript. After reading it, it looks like the novelty only comes from the experimental section, since the use of BoF and TCR regularization has already been proposed in other papers, like the version of (Chen et al. 2022a) being [reviewed for TMLR](https://openreview.net/forum?id=r06xREo3QG&referrer=%5BTMLR%5D(%2Fgroup%3Fid%3DTMLR)).

On the results:

* Throughout the manuscript, the comparison with other methods is done at epoch level. While this is interesting from an information-theoretic point of view (although not explored in this work) this work focuses more on the practicality and efficiency of SSL. If that is the case, I believe the comparison should be done in terms of time (as done in Table 3) or in terms of the total number of patches processed by the backbone. Since one BYOL epochs requires $2N$ patches to be processed, while EMP-SSL requires $200\times 2N$. In my opinion, this would give ML practitioners a more objective idea of the practicality of EMP-SSL vs other established methods.
  * For example, In Table 3 it is shown that EMP-SSL is faster. However, EMP-SSL @ 200 patches requires 30 epochs to reach SOTA. This amounts to $142 / 5 * 30 = 852$. Conversely, BYOL requires 1000 epochs to reach SOTA, amounting to $142/310*1000 = 458$, which is much smaller than $852$. I think it would be interesting to show the actual time required to reach SOTA, since intermediate dynamics (eg. reaching 90%) are very different among methods, and of less interest for a final application.

* What is the efficiency of EMP-SSL at test time? As far as I understand, one would require to run the backbone over K patches, and then aggregate the features. How does this compare to 1 backbone run in BYOL for example? Test efficiency is of great interest for applicability in real product, I think discussion in that sense should be added.


* Is SwAV implemented with multicrop? It is known that SwAV specifically improves by an important margin when using multicrop. In general, how does EMP-SSL compare to some intense version of multi-crop, say with 20 or 200 patches per image? I think the main comparison in this work should be with multicrop, since it is a closer strategy to the proposed one than 2-view methods.

* While C10 and C100 are interesting, I think these datasets are extremely small to extract broad conclusions about SSL. The minimal size I think can generalize is ImageNet-1K. Additionally, the cropping strategy might behave much differently for 224x224 images than for 32x32 ones. Some discussion, and ideally experiments, in that sense would be very valuable.


On the mathematical notation:

* I would mention that $D(Z_1,Z_2) = Tr(Z_1^\top Z_2)$ is inefficient, since only the diagonal elements are collected after all. Is this implemented efficiently in practice?

* I suggest removing $Z = [Z_1,\ldots , Z_n]$ since it is not used later on and creates confusion with Eq.1, which is used only with $Z_i$ anyway (in Eq.2).

------
General comment:

I am willing to increase my score upon fruitful discussion. However, as the paper is, there is a lack of clarity about the novely. Furthermore, the comparison in terms of "epochs" sounds unfair from my point of view, being wall-time or FLOPS more appropriate. The datasets used are too small for broad conclusions, and further discussion is required (test efficiency, gpu memory, etc.).

---

### Official Review · Reviewer_tgxj · 2023-10-28

**Soundness:** 3 good
**Presentation:** 3 good
**Contribution:** 3 good
**Rating:** 5
**Confidence:** 4

**Summary:**

This paper proposes a new method that aims to improve the efficiency of contrastive learning methods for image representation learning. The main idea is to create a large number of patches from every single image, and then use a VicReg style loss function to learn representation, such that patches from the same source image have similar representations in the sense of cosine similarity, and patches from different images have distant representations. The representation of an image is then the average of the representations of all patches from the same image. The authors run experiments and focus on the regime where the number of training epochs is small. With even one epoch of training, they show that this method achieve arguable non-trivial linear probe accuracy on datasets such as CIFAR and ImageNet-100, whereas more classic methods like SimCLR requires many more epochs.

**Strengths:**

- The paper is well-written, the definition of the method is quite clear and easy to understand.
- The authors spent extensive efforts trying to do ablation study, including linear probe accuracy at different number of epoch and with different number of patches.
- There's also discussion about the time complexity which to me sounds essential for this work to be convincing.

**Weaknesses:**

- In most of the experiments, the accuracy number of the proposed method is much lower than that of the baseline methods. Although I understand that the main point of the paper is to say that this method can converge faster, I feel this conclusion is irrelevant when the final accuracy is off by a lot. For example, in Table 1, the accuracy of the proposed methods is at least 5% lower on any dataset, and even 20% lower on the most realistic one (ImageNet-100). This makes this table kind of useless in my opinion.
- The authors don't provide a clean table comparing the wall-clock time spent to achieve the same SOTA accuracy using their method versus other baseline methods. This seems to me the most important thing to include, because the definition of 'epoch' in this paper is kind of irrelevant given that many more augmentations are used in each epoch in their method. Only the wall-clock time seems to be really reasonable metric to show that this method is faster.
- Despite trying to include a little bit of result on ImageNet-100, most results (especially those that are used to argue the wall-clock time improvement) are based on CIFAR which is too small in modern deep learning.

Given the current empirical results (which is not completely convincing to me) and the nature of the method (not very noval given that it's just using more patches wih VicReg), I don't think it's ready to be published yet unless more convincing details are included.

**Questions:**

What happens if you just tune the hyperparmeters for SimCLR or VicReg to make them converge faster in the initial epochs? I think the current lr schedule is designg to optimize final performance, but if the goal is really the convergence efficiency, it sounds to me that there could exist much simpler ways to achieve that without much modification of the algorithm.

---

### Official Review · Reviewer_FQ2Z · 2023-10-31

**Soundness:** 2 fair
**Presentation:** 3 good
**Contribution:** 2 fair
**Rating:** 5
**Confidence:** 4

**Summary:**

This paper proposes a self-supervised learning method called EMP-SSL, which can converge and achieve good performance within a few epochs. The main technique used in the proposed method is to augment the images into multiple views and ensure them to be semantically closed. Additional regularizer is added to avoid collapsed representation. The experiment results suggest that the proposed EMP-SSL can reach a relatively good result within only 1 epoch on various standard datasets. The method can also reach better results with more epochs of training. The good transferability to OOD data is also promised by experiments.

**Strengths:**

* S1: The proposed method is indeed simplistic as the authors have claimed, which can be easily implemented.
* S2: The experiment settings and datasets are variously chosen, which ensures the generality of the conclusion to some extent.
* S3: The results of pretraining only 1 epoch are surprising.

**Weaknesses:**

* W1: The so-called one epoch is a little tricky, since EMP-SSL use 20-200 views for one image, which is far more than conventional methods have used. EMP-SSL will use at least 10 times of the data as much as conventional SSL methods use in one epoch.
* W2: I do not see results with ResNet-50 network, which may be a more standard setting for self-supervised learning. Some conclusions reached with smaller networks may not be generalized to larger networks.
* W3: While I approve of the note on time efficiency, the comparison of time efficiency is not intuitive. I still wonder how many epochs can be run using conventional methods like SimCLR in the time of running 1 epoch of EMP-SSL.
* W4: The authors mentioned co-occurrence in their motivation of the methods. However, I have no idea what the co-occurrence is represented here and how this technical term is defined.
* W5: The paper may lack some ablation study on the choice of the loss function. Given 200 views of one image, I believe there are more than one ways to design the loss function and the regularizer, i.e. InfoNCE loss.

**Questions:**

Besides the questions raised in the weakness part, I have several additional questions:
* Q1: I notice that the patch cropped from the original image is in a fixed size. Is there any motivation for that this size is chosen and that the size is fixed. How about using a random size just like the cropping operation in most self-supervised learning method?
* Q2: The experiments support the fact that EMP-SSL is good at transfer learning. Can the authors provide an intuitive explanation?
* Q3: The authors use bag-of-feature representation, which will produce different results in each evaluation. What is the variance of these results?

---

### Official Review · Reviewer_tvSj · 2023-11-01

**Soundness:** 2 fair
**Presentation:** 2 fair
**Contribution:** 3 good
**Rating:** 3
**Confidence:** 4

**Summary:**

In this paper the authors propose a new solution, EMP-SSL, a self-supervised learning method able to converge the in just one training epoch. The authors compared their method with other SSL methods on 4 different datasets.

**Strengths:**

- The authors extensively tested their method on four datasets
- The authors presented two ablations studies for assessing their method.
- The idea of doing SSL in just one epoch is very interesting and promising for the community
- The authors compared with a large number of SSL methods.

**Weaknesses:**

While the authors present an innovative method, it appears that the paper lacks experiments to analyze the effectiveness of the method, and the results are still not convincing. Here are a few specific points of concern:

- The literature suggests that SSL methods benefit from larger batch sizes (for instance, SimCLR uses 4096). However, the batch sizes considered by the authors range from 0 to 32 (including the ablation study in the supplementary material). This significant variation in the number of batch iterations and weight updates makes a fair comparison with other methods difficult. Conducting experiments with a consistent batch size would help determine if these methods, with a large number of updates, can achieve reasonable performance. Consequently, the results in tables 1 and 2 lack credibility.
- The out-of-domain data experiment presented in table 5, involving cifar10 and cifar100, lacks depth. While the left part of the table might be suitable for testing the model's out-of-domain capability, the right part lacks usefulness. A more insightful approach would involve testing the model on a completely different dataset to better assess its transferability.
- The paper overlooks experiments related to downstream tasks, which are crucial in SSL for evaluating the quality of features learned by the model.
- The ablations and experiments conducted on cifar10 lack informativeness. A larger dataset could have been utilized for more robust analysis.
- The paper's narration needs enhancement, particularly on page 2, where method details are repetitively presented, leading to confusing notations.
- The paper lacks a hyperparameter analysis, including sensitivity analysis concerning parameters such as learning rate and lambda in the TCR loss function.

**Questions:**

- Why did the authors not present an architecture analysis? Perhaps different architectures could enhance the model's performance.
- A comparison of GFlops, maintaining the same batch size, with other methods could be useful.
- Please provide more details for the T-SNE analysis, as the existing information is not convincing.
- Why did the authors limit the analysis of the batch size to 32? Please provide results with increased dimensions.